# Development of a core outcome set and identification of patient-reportable outcomes for primary brain tumour trials: protocol for the COBra study

Ameeta Retzer [1,2] Stephanie Sivell,[3] Hannah Scott,[4] Annmarie Nelson,[3] Helen Bulbeck,[5] Kathy Seddon,[6] Robin Grant,[7] Richard Adams,[8] Colin Watts [9] Olalekan Lee Aiyegbusi [1,10] Pamela Kearns,[9,10] Samantha Cruz Rivera [1,11] Linda Dirven,[12,13] Elin Baddeley [3] Melanie Calvert [1,2] Anthony Byrne[3]

For numbered affiliations see end of article.

**Correspondence to**
Dr Anthony Byrne;
Anthony.Byrne2@wales.nhs.uk

## ABSTRACT

**Introduction** Primary brain tumours, specifically gliomas, are a rare disease group. The disease and treatment negatively impacts on patients and those close to them. The high rates of physical and cognitive morbidity differ from other cancers causing reduced health-related quality of life. Glioma trials using outcomes that allow holistic analysis of treatment benefits and risks enable informed care decisions. Currently, outcome assessment in glioma trials is inconsistent, hindering evidence synthesis. A core outcome set (COS) - an agreed minimum set of outcomes to be measured and reported - may address this. International initiatives focus on defining core outcomes assessments across brain tumour types. This protocol describes the development of a COS involving UK stakeholders for use in glioma trials, applicable across glioma types, with provision to identify subsets as required. Due to stakeholder interest in data reported from the patient perspective, outcomes from the COS that can be patient-reported will be identified.

**Methods and analysis** Stage I: (1) trial registry review to identify outcomes collected in glioma trials and (2) systematic review of qualitative literature exploring glioma patient and key stakeholder research priorities. Stage II: semi-structured interviews with glioma patients and caregivers. Outcome lists will be generated from stages I and II. Stage III: study team will remove duplicate items from the outcome lists and ensure accessible terminology for inclusion in the Delphi survey. Stage IV: a two-round Delphi process whereby the outcomes will be rated by key stakeholders. Stage V: a consensus meeting where participants will finalise the COS. The study team will identify the COS outcomes that can be patient-reported. Further research is needed to match patient-reported outcomes to available measures.

**Ethics and dissemination** Ethical approval was obtained (REF SMREC 21/59, Cardiff University School of Medicine Research Ethics Committee). Study findings will be disseminated widely through conferences and journal publication. The final COS will be adopted and promoted by patient and carer groups and its use by funders encouraged.

**PROSPERO registration number** CRD42021236979.

## STRENGTHS AND LIMITATIONS OF THIS STUDY

⇒ This study collects original qualitative data to ensure all outcomes prioritised by glioma patients are identified. However, this is a resource-intensive process that may not be available to all core outcome set developers.

⇒ Review of trial registries represents a pragmatic approach to comprehensively identify outcomes used in trials rather than reliance on often incomplete outcome reporting in glioma trial publications. There are limitations to this approach—use of trial registries means those that are not registered will not be identified, registry use is inconsistent globally, completeness and specificity can be questionable, and updating of entries continues to be a challenge. However, the quality of registration has been observed to be improving and trial registration associated with subsequent publication and use of the same outcomes as defined in their protocols as in their published reports.

⇒ Bias may be introduced by inviting qualitative interview participants from stage II to take part in the Delphi; though this encourages familiarity with concepts, enabling meaningful participation in the Delphi.

⇒ Qualitative data collected from the UK population may limit international applicability, though this allows exploration of issues that may be specific to UK context and validation of this core outcome set for use in other settings should be explored.

## INTRODUCTION

Primary brain tumours, specifically gliomas, are part of a rare disease group.[1] The disease and its treatment have negative effects on patients and those close to them. The high rates of physical and cognitive morbidity differ from other cancers, with significant impact on a wide range of functional domains. Gliomas are the most common form of primary brain tumour,[2] accounting for 80% of malignant

brain tumours. Gliomas represent a heterogeneous group of cancers with variable outcome, traditionally graded from I to IV (least to most aggressive). However, rapid developments in molecular diagnostics have led to refinements in nomenclature, suggesting a more nuanced approach to brain tumours classification.[3] This would acknowledge the spectrum ranging from a variable but slower-progressing course, such as oligodendroglioma or astrocytoma, to fast-growing tumours such as glioblastoma, a particularly aggressive subtype with a median survival of 12–15 months and 5% five-year survival rate.[4]

The poor prognosis of some glioma patients and the high symptom burden has led to a growing emphasis on their quality of survival.[5] Maintaining cognitive function, physical function and other health-related quality of life aspects throughout the disease trajectory are key considerations alongside very modest survival benefits captured through traditional metrics of tumour response and overall or progression-free survival, particularly for patients with aggressive forms of glioma.[6] Therefore, it is important that glioma intervention studies collect a range of data aligned with patient priorities to enable assessment of the net clinical benefit of treatments.[7–10]

Data collected to evidence effects of interventions are known as 'outcomes'. Outcomes include traditional measures such as progression-free survival and radiological tumour response but also Clinical Outcome Assessments (COAs). COAs describe how a patient feels, functions or survives. COAs include Clinician Reported Outcomes, Observer Reported Outcomes, Performance Outcomes and Patient Reported Outcomes (PROs).[11] PROs assess a range of outcomes including symptoms, functional health, well-being and psychological issues from the patients' perspective, without interpretation by a clinician or anyone else.[12] When assessing treatments, PROs enable insight into the impact of treatment on patient's perceived well-being where other outcome data that may indicate minimal differences in disease control and survival, potentially influencing patients' treatment choices.[13]

Interpreting the clinical benefit of treatments requires effective data synthesis and meta-analyses of trial outcomes. This requires consistent use of outcomes, use of appropriate outcome measures, and diligent data capture, analysis and reporting. Inconsistent outcome use is widespread. A significant lack of standard ontology has been found in cancer clinical trials[14] and in brain tumour studies specifically.[15] Moreover, selective outcome and missing data reporting is common,[16] introducing bias and hindering evidence synthesis. PROs are critical to the comprehensive evaluation of treatment benefits and side effects, and are increasingly used by regulatory authorities. The Food and Drug Administration (FDA) is prioritising a patient-centred approach to drug development,[17] a consistent approach to PRO use generally[18] and in cancer clinical trials specifically.[19] The European Medicines Agency (EMA) support PRO use to assess drug efficacy and tolerability in informing product approval in cancer,[20] consistent with the FDA.[21 22] Key PROs for use

in cancer has been of consistent interest,[19 23 24] patients value this form of data[25–28] and it underpins informed shared decision-making.[29–32] However, there is a limited consensus on which areas of patient experience should be consistently assessed in brain tumour trials. In cancer trials using PROs, analyses are often unreported in publications and the clinical relevance of PRO results are overlooked.[33] A systematic review of glioma randomised controlled trials using PROs found that only 14% of these trials met the criteria for high-quality reporting,[34] with PRO results not being interpreted in 79%, and clinical relevance not discussed in 86% of trials.

There are international efforts to unify and improve practice. In PRO research in the field of neuro-oncology, the Response Assessment in Neuro-Oncology Patient Reported Outcomes (RANO-PRO) working group aims to provide guidance on Patient-Reported Outcome Measures in adult neuro-oncology clinical trials and practice. Their systematic review[15] found that 215 PROs have been used in brain tumour (primary and secondary) studies, the majority only used once or twice. The FDA and EMA recognise the importance of assessing symptoms, adverse effects and function as core constructs in all glioma trials,[35] and have participated in an international multi-stakeholder workshop aiming to define a core set of priority constructs to be assessed as minimum in high-grade glioma trials and care.[36]

Core outcome sets (COS) establish 'the minimum that should be measured and reported in all clinical trials of a specific condition',[37] aiming to achieve consensus between researchers, clinicians, patients and policy makers. This facilitates consistent outcome collection, analysis and reporting, enables data synthesis and meta-analyses, reduces research waste and informs patient-centred care. On COS confirmation, further research will determine how to measure these outcomes.

The primary aim of this research is to develop a COS for use in adult primary glioma (astrocytoma, oligodendroglioma, oligoastrocytoma, ependymoma, astroblastoma, anaplastic ganglioglioma, glioblastoma, glioblastoma multiforme) phase III interventional trials comprising all outcome types. We will define outcomes applicable to all glioma as well those that may be specific to glioma types. The COS will inform interpretation of the net clinical benefit of interventions in terms that reflect stakeholder priorities. Due to interest in core PROs in cancer, our secondary aim is to identify the COS outcomes which can be patient reported.

## Focus of COS

This COS will apply to phase III interventional trials for systemic anticancer treatments (including immunotherapy and chemotherapy), radiotherapy, surgery and supportive care involving adults (aged over 18 years), diagnosed with glioma, with a specific focus on the UK population. Though some data formulating this COS will be drawn from a UK sample, trialists should consider the COS to be applicable internationally. To promote generalisability of results, recruitment into the qualitative

interviews and Delphi exercise will be monitored for glioma type, age, ethnicity and gender.

## METHODS AND ANALYSIS
### Objectives
1. Trial registry review to identify glioma trial outcomes and a systematic review of the qualitative literature to explore key stakeholders' research and treatment priorities.
2. Identify outcomes using qualitative interviews with glioma patients and caregivers.
3. Combine the results of objectives 1 and 2 into a unified longlist of outcomes.
4. Achieve consensus on a COS through online Delphi process and a consensus meeting with a range of stakeholders.

### Study design
The COBra (Patient Reported Core Outcomes in Brain Tumour Trials) study uses a mixed-methods, multi-stage approach in accordance with accepted COS methodology[38] and guidance[39] (online supplemental appendix 1) and registered with the Core Outcome Measures in Effectiveness Trials (COMET) initiative.[40]

### Study team and collaborators
The study team is multidisciplinary, including Patient and Public Involvement (PPI) representatives, healthcare professionals, researchers, policy makers, and regulators.

The Marie Curie Palliative Care Research Centre Cardiff, the Centre for Patient Reported Outcomes Research, the Centre for Trials Research (Cardiff University) and Birmingham Clinical Trials Unit (CTU) will provide methodological steer on behalf of the Supportive and Palliative Care subgroup of the NCRI (National Cancer Research Institute) Brain Tumour group. Collaboration with the RANO-PRO initiative[41] working group will ensure alignment with international efforts.

### Patient and public involvement
The PPI team members contributed to the study design and will develop and monitor the study as part of the Steering Group,[42] contributing to data analysis and dissemination of study findings. The study team will seek advice from a wider panel of PPI representatives convened for the purpose of the study, consisting of individuals with a range of backgrounds and experiences. The detailed participation of PPI representatives will be reported in accordance with Guidance for Reporting Involvement of Patients and the Public.[43]

### Stage I: evidence review
#### Aims
Review of clinical trial registries and a systematic review of published qualitative literature to generate an outcome list[38] from:
A. Phase III interventional glioma trials involving adult patients and diagnosed with primary glioma.

B. Qualitative studies exploring the lived experience and research priorities of adult patients with primary glioma, and other key stakeholders.

### Search strategy and data extraction
#### Search A
ClinicalTrials.gov and ISRCTN clinical trials registries, based in the USA and UK, respectively, will be used to identify outcomes used in phase III interventional glioma trials in adults (online supplemental appendix 2). Data from both are available for public download. Where protocols are available alongside registration information, these will be retrieved.

Two reviewers will independently perform complete searches of glioma trials registered on clinicaltrials.gov and isrctn.com without restriction by date. The results will be independently reviewed for eligibility; disagreements will be resolved with a third reviewer. Two reviewers will independently extract data including basic trial information, year of study, primary outcome(s) and secondary outcomes. Data in the csv files will be cross-referenced with clinical trial registration entry for completeness, and with the protocol when available. The most recently updated of these will be used.

Trials sourced during Search A will be cross-referenced with those retrieved from the RANO-PRO study for information.

#### Search B
We will systematically review the qualitative literature describing the experiences and needs of adults diagnosed with glioma and thematically synthesise[43] their 'lived experiences' in relation to care, treatment and treatment outcomes.

Databases to be searched include MEDLINE, EMBASE, CINAHL, Web of Science, PsycINFO, the Cochrane Central Register of Controlled Trials and the Cochrane Library. Reference lists of key authors and journals will be hand searched. Qualitative studies or mixed-method studies containing qualitative data, published in the English language, restricted to 15 years prior, will be included. This is because of limited data prior and literature captured is more reflective of current treatment options and patient perspective Research involving adult patients and/or key stakeholders including informal carergivers, will be included. Two reviewers will independently review all titles and abstracts; a third reviewer will review citations for any disagreements. Full text studies will be reviewed by two reviewers; disagreements will be resolved with a third reviewer.

Two reviewers will independently extract data using a standardised data collection form, capturing the themes and sub-themes of the qualitative data pertaining to the lived experience of patients with primary glioma. The qualitative literature will be thematically synthesised following three stages: coding text, developing descriptive themes and generating themes.[44] The data will focus on patients and key stakeholders including informal

carergivers, exploring their interpretation of patients' 'lived experiences', including views relating to their attitudes and experience of symptoms and functional outcomes. NVivo[45] will be used for data management.

## Stage II: interviews with patients and caregivers

Semi-structured interviews will be conducted with adults diagnosed with primary glioma across the spectrum of the disease. Interview participants can identify a caregiver to join them in an interview dyad. The interviews will inform the language used in the Delphi survey and identify outcomes not captured during stage I.

### Aims

The objectives of these interviews are to explore:
► Outcomes that are important to patients.
► Caregivers' understanding of patients' priorities and experiences, as these may differ.

### Participant eligibility and sampling

Dyads will comprise eligible patients histologically diagnosed with primary glioma (astrocytoma, oligodendroglioma, oligoastrocytoma, ependymoma, astroblastoma, anaplastic ganglioglioma, glioblastoma, glioblastoma multiforme) and a caregiver identified by the patient. Caregivers are defined as informal carers, who may be a family member or friend, who provides the majority of the support to the patient and is able to estimate the patient's priorities. Patients and caregivers will be over the age of 18 years.

Participants will be recruited through the NCRI Brain Group, the Tessa Jowell BRAIN MATRIX trial platform,[46] CTUs, brainstrust—the brain cancer people, The Brain Tumour Charity, snowballing, known contacts and social media platforms. Potential participants will be invited to contact the research team to express interest. Recruitment will be monitored to promote diversity in terms of glioma type, age, ethnicity and gender, seeking balance between glioma types. Between 12 and 20 dyads representing the spectrum of malignant disease will be recruited based on previous studies and expected data saturation.[47] Data saturation will be assessed through constant discussion and evaluation of the data by the qualitative researchers conducting the data collection and analysis, together with members of the wider study team. Recruitment will end when data saturation is reached.

### Consent and capacity

Patients and caregivers will give consent on their own behalf if they wish to participate in an interview. If a patient or caregiver does not proceed with an interview, the other will still be invited to participate. Their permission is not required for the other to participate. Information sheets will be sent to eligible participants via post or email with the contact details of the research team member conducting the interviews. Participants expressing interest will be given the chance to ask any questions prior to consent. Participants will complete an electronic or hardcopy consent form or will be recorded giving verbal consent, depending on interview format.

In accordance with the Mental Capacity Act (2005), patient participants will be assumed to have capacity unless it is proven otherwise. If there is concern that the patient lacks capacity to participate, this will be discussed with the Chief Investigator, a clinician, about whether further research activity will occur. If research will not continue with the patient participant, the caregiver will be given the opportunity to take part in an interview to share their views.

### Data collection

A semi-structured interview format will be used to understand patient experiences of living with glioma, and what they consider to be the most important outcomes from glioma treatment. Caregiver participants' perspective of patients' experience and priorities will be captured, not a direct report of the patients' condition. The interviews will be undertaken via phone or video link (eg, Zoom or Microsoft Teams), or face-to-face, depending on the situation and preference of patients. Interviews may take place with patients and caregivers together or separately, depending on their preference. Interviews where patients and caregivers are interviewed separately allow for differing views to be expressed. Where interviews are undertaken together, efforts will be made to ensure both are able to express their views. Interviews will be audio-recorded. The interview will be guided by open-ended questions on diagnosis, treatment and their effects on patients and caregivers, directed towards understanding outcomes important to patients. The semi-structured format allows for spontaneous exploration of novel topics. The topic-guide may be reviewed and adapted iteratively after the first few interviews, if required. At the end of the interviews, participants will be asked directly which outcomes they believe should be measured in clinical trials. This places the lived experience of participants at the forefront, with patients and caregivers given the chance to talk about the things that matter most to them.

### Data analysis

The interview data, once transcribed and anonymised, will be thematically analysed[48] using NVivo software[45] for data management. A preliminary framework will be derived from the available literature including the Thematic analysis allows for the identification of patterns and themes within the data, to organise and describe data in rich detail.[48] It is particularly well-suited to studies that focus on lived experience. Data collected from patients and caregivers will be analysed and formulated into separate accounts.

Analysis of the first three transcripts will be conducted independently by two members of the research team experienced in qualitative research and a draft coding structure will be formulated. Disagreements in coding will be resolved through discussion and input from a third qualitative researcher will be sought when required. The

draft coding frame will be reviewed by PPI team members and a coding structure for the remaining transcripts will be confirmed. The framework will be refined, until the analysis of all transcripts has been completed, with the findings synthesised into categories and subcategories.

## Stage III: review of outcome list

All outcomes, without limitation by outcome type, captured in stage I will be grouped and classified.[38] A broad ontology for this will be developed from the framework outlined in the COMET handbook and relevant frameworks from the available literature[35] in advance of outcome extraction and will be iteratively refined based on the outcomes identified. The ontology will serve as a categorical tool to organise and present the outcomes in an accessible manner. Each grouping will contain domains and subdomains that broadly measure particular aspects of the effects of interventions (eg, symptoms and function).[49] The outcome lists formed by each of the two researchers will be compared for completeness, and differences in the categorisation will be resolved through discussion.

The categories and subcategories generated in stage II will be formulated into an outcome list and differences in the categorisation will be resolved through discussion.

A longlist of outcomes will be generated from the stage I and II outcome lists. Duplicates will be removed during this process. This list will be reviewed by the study team to refine the language used to describe the outcomes. The team will review the structure of the questions included in the Delphi survey. At this stage, it will be decided whether separate Delphi processes are needed according to glioma type based on the emerging data.

## Stage IV: Delphi survey

A modified two-round Delphi will be used to assess the relative importance of outcomes included in the stage III outcome list. Participants will be invited to consider applicability of the COS to new and emerging therapies, and whether the outcomes would apply. The aim of the Delphi process is to reach consensus on which outcomes should form the COS for glioma trials.

### Recruitment

Approximately 100 participants with professional or personal experience of glioma care and treatment: (1) patients, (2) caregivers, (3) healthcare professionals and researchers, (4) policy-makers and regulators will be recruited as previously described in earlier stages. During Delphi registration participants will choose the stakeholder group with which they most identify but can note if they identify with other stakeholder groups besides their primary. Approximately 25 participants will be recruited to each stakeholder group, recruitment will be monitored and will inform and direct efforts as required. Consent will be taken electronically during the online registration process.

### Delphi process

The Delphi exercise will reflect COMET recommendations[38] and will present the stage III outcome list. Participants will rate each of the outcomes on a 9-point Likert scale, (1–3, not important; 4–6, important but not critical; and 7–9, important and critical).[50] During round 1, participants can add outcomes they feel are missing. Votes from individuals in each stakeholder group will be given equal weighting. All original outcomes will be presented in round 2. Outcomes added by participants in round 1 will be presented in round 2. In round 2, respondents will be presented with their own rating for each outcome and how it was rated by their own stakeholder group. Based on this information, respondents will be invited to amend their score, if they wish. During round 2, participants can rate the outcomes suggested in round 1.

The threshold for consensus for inclusion in or exclusion from the COS will be ≥70%, informed by those used in comparable COS development studies.[51 52] After the Delphi, outcomes will be proposed for inclusion in the final COS if ≥70% respondents rate the item as 7–9 and ≤15% rate the item as 1–3. Items will be proposed for exclusion from the final COS if ≥70% respondents rate the item as 1–3 and ≤15% rate the item as 7–9. Those outcomes that do not reach agreement after the two Delphi rounds will be discussed in the consensus meeting, together with the items proposed for inclusion and exclusion.

### Missing data

To minimise partial response, participants will be unable to skip questions but can indicate when they feel unable to rank specific items. Reminders will be used to minimise participant attrition between Delphi rounds. Use of specialised Delphi software, Delphi Manager, will enable rapid inter-round rating calculations to allow the second round to open with minimal delay to further reduce attrition.

## Stage V: consensus meeting

This meeting may be held virtually or in person, depending on the situation and preference of the majority of participants. All Delphi participants will be invited. Notes will be taken during the meeting and consent will be sought from all participants to audio-record the meeting for reference. Decisions made during the consensus meeting will be made through anonymous voting using voting software. Decisions will proceed if ratified by ≥70% of the group. In cases where there is <100% consensus, decisions will be discussed until those in disagreement are satisfied that their views have been considered and that the decision can proceed. This meeting allows for a further opportunity to discuss, validate and confirm the final COS. The core outcomes applicable to all glioma trials will be agreed, as will any outcomes identified as specific to particular types of glioma. Following the consensus meeting, the study team will identify which of the outcomes could be assessed by patient reporting.

## Ethics and dissemination

Ethical approval was granted (REF: SMREC 21/59, Cardiff University School of Medicine Research Ethics Committee). All data will be collected and stored in accordance with local regulations.[53]

The final COS will be published in compliance with accepted reporting standards[38] and adopted and promoted by the NCRI Brain Clinical Studies Group Supportive and Palliative Care subgroup for use in glioma studies. The subgroup will publish a position statement mandating for UK CTUs involved in brain tumour research to implement the COS.

Study findings will be disseminated widely, including to national and international conferences and high-impact journals. A plain English summary will be coproduced with PPI team members and made available to participants on request. The COS will be promoted among patient and carer groups using The Brain Tumour Charity network (including BRIAN), NCRI and regional PPI frameworks, brainstrust and other patient organisations. The importance of COS development is increasingly recognised by funders, such as the National Institute for Health and Care Research, and regulators, such as EMA and FDA. The COS will therefore be promoted to encourage its inclusion in 'justification of outcomes' sections of funding proposals and regulatory submissions. The final COS will be freely available on the COMET database.

Though the participants in the original qualitative data collection will be drawn from a UK sample and the Delphi participants will be largely based in the UK, the trial registry searches were without restriction based on country and the qualitative literature was limited to those in English language only. The study steering committee has membership from stakeholders leading international initiatives and the Delphi survey and consensus meeting will involve participants from international regulatory bodies. As a result, the resulting COS should be considered to be internationally applicable. For use in other settings or countries, validation exercises are advised to ensure economic and cultural differences are integrated. The study team will consider the findings of this study in the context of existing international initiatives. Findings will be shared with international partners and may be integrated into international guidance on outcome assessment across all brain tumour types.

COBra will directly collaborate with the RANO-PRO working group and affiliated international initiatives. Following study completion, RANO-PRO findings may be used to select appropriate COAs aligned to the COS. COBra will also collaborate with UK funders, trialists and CTUs on COS implementation and the consistent application of international standards for collection, analysis and reporting of the COS across all UK studies.

Following finalising the COS, further research is required to identify and/or develop corresponding outcome measures.

**Author affiliations**
[1]Centre for Patient Reported Outcomes Research, Institute for Applied Health Research, University of Birmingham, Birmingham, UK
[2]National Institute for Health and Care Research (NIHR) Applied Research Centre, West Midlands, Birmingham, UK
[3]Marie Curie Palliative Care Research Centre, Division of Population Medicine, Cardiff University School of Medicine, College of Biomedical and Life Sciences, Cardiff University, Cardiff, UK
[4]Cambridge Public Health, University of Cambridge School of Clinical Medicine, University of Cambridge, Cambridge, UK
[5]brainstrust, Cowes, UK
[6]Cardiff University, Cardiff, UK
[7]Department of Clinical Neurosciences, Royal Infirmary of Edinburgh, Edinburgh, UK
[8]Centre for Trials Research, Cardiff University, Cardiff, UK
[9]Institute of Cancer and Genomic Sciences, College of Medical and Dental Sciences, University of Birmingham, Birmingham, UK
[10]NIHR Birmingham Biomedical Research Centre, University of Birmingham, Birmingham, UK
[11]Birmingham Health Partners Centre for Regulatory Science and Innovation, University of Birmingham, Birmingham, UK
[12]Department of Neurology, Leiden University Medical Center, Leiden University, Leiden, The Netherlands
[13]Department of Neurology, Medical Centre Haaglanden, Den Haag, The Netherlands

**Contributors** The study concept and design was conceived by AR, SS, HS, AN, HB, KS, RG, RA, CW, OLA, PK, SCR, LD, EB, MC and AB. MC and AR advised on methodology. EB and AR will undertake the registry review, EB and SS will undertake the qualitative systematic review. EB and SS will recruit, screen and consent participants and will undertake the interviews with input from AR and AB. EB will recruit for the Delphi and consensus meeting, with input from SS, AR and AB. HS prepared the first draft of the manuscript. AR prepared subsequent drafts. SS, HS, AN, HB, KS, RG, RA, CW, OLA, PK, SCR, LD, EB, MC and AB all provided edits and critiqued the manuscript for intellectual content.

**Funding** This study is funded by The Brain Tumour Charity (GN-000704). The funders have no role in study design or manuscript preparation.

**Competing interests** Due to their involvement in the study design, the study team members will not participate in the Delphi process or consensus meeting, other than in a facilitative role. Study team members will encourage engagement and participation in the Delphi process and consensus meeting by individuals within the networks of which they are part, as appropriate. AR is partially funded by NIHR Applied Research Collaborative West Midlands. SS is supported by Marie Curie core grant funding to the Marie Curie Palliative Care Research Centre, Cardiff University, grant reference MCC-FCO-11-C. MC is Director of the Birmingham Health Partners Centre for Regulatory Science and Innovation, Director of the Centre for Patient Reported Outcomes Research and is a National Institute for Health Research (NIHR) Senior Investigator. She receives funding from the NIHR Birmingham Biomedical Research Centre, the NIHR Surgical Reconstruction and Microbiology Research Centre and NIHR ARC West Midlands at the at the University of Birmingham and University Hospitals Birmingham NHS Foundation Trust, Health Data Research UK, Innovate UK (part of UK Research and Innovation), Macmillan Cancer Support, UCB and GSK Pharma. MC has received personal fees from Astellas, Takeda, Merck, Daiichi Sankyo, Glaukos, GSK, Aparito Ltd, CIS Oncology and the Patient-Centered Outcomes Research Institute (PCORI) outside the submitted work. MC has led the development of SPIRIT-PRO and CONSORT-PRO international guidance and is a member of the SISAQOL and PROTEUS Consortia. OLA receives funding from the NIHR Birmingham Biomedical Research Centre (BRC), NIHR Applied Research Centre (ARC), West Midlands at the University of Birmingham and University Hospitals Birmingham NHS Foundation, Innovate UK (part of UK Research and Innovation), Gilead Sciences Ltd and Janssen Pharmaceuticals, Inc. OLA declares personal fees from Gilead Sciences Ltd, GlaxoSmithKline (GSK) and Merck outside the submitted work. LD has received research funding (as PI or collaborator) from the Brain Tumour Charity, EORTC Quality of Life Group, Innovative Medicines Initiatives, Dutch Cancer Society, ZonMw, National Institute for Health Research UK and crowd funding initiatives, although not related to this specific research. LD is involved in the EORTC (as chair of the Brain Tumour Group Quality of Life Committee) and in the development of EORTC questionnaires for brain tumour patients (IADL, BN20). LD is a representative of the RANO-PRO initiative. LD is associate Editor for Neuro Oncology, dealing with papers on clinical outcomes in brain tumours.

**Patient and public involvement** Patients and/or the public were involved in the design, or conduct, or reporting, or dissemination plans of this research. Refer to the Methods section for further details.

**Patient consent for publication** Not applicable.

**Provenance and peer review** Not commissioned; externally peer reviewed.

**ORCID iDs**
Ameeta Retzer http://orcid.org/0000-0002-4156-8386
Colin Watts http://orcid.org/0000-0003-3531-8791
Olalekan Lee Aiyegbusi http://orcid.org/0000-0001-9122-8251
Samantha Cruz Rivera http://orcid.org/0000-0002-1566-6804
Elin Baddeley http://orcid.org/0000-0002-7571-4820
Melanie Calvert http://orcid.org/0000-0002-1856-837X

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
