## [Reviewer comments · BMJ Open]

ARTICLE DETAILS

TITLE (PROVISIONAL)	Development of a Core Outcome Set and Identification of Patient-Reportable Outcomes for Primary Brain Tumour Trials: Protocol for the COBra Study
AUTHORS	Retzer, Ameeta; Sivell, Stephanie; Scott, Hannah; Nelson, Annmarie; Bulbeck, Helen; Seddon, Kathy; Grant, Robin; Adams, Richard; Watts, Colin; Aiyegbusi, Olalekan Lee; Kearns, Pamela; Cruz Rivera, Samantha; Dirven, Linda; Baddeley, Elin; Calvert, Melanie; Byrne, Anthony

VERSION 1 – REVIEW

REVIEWER	Bhagavatula Indira Devi National Institute of Mental Health and Neuro Sciences
REVIEW RETURNED	19-Jan-2022

GENERAL COMMENTS	Glioma outcome trials should have perspective from all the stakeholders . a comprehensive outcome measure would give a better choice of selection and planning for all concerned . this kind of study to measure outcome in a comprehensive way is welcome . i suggest the survey questions should be comprehensive and should not be just question number 123 etc but stakeholder should be able to answer q no 8 . there should be provision for missing data which the participant can answer separately . many of the questionnaires fall short on this when the participant does not attempt an answer or the answer is not any of the options . this can be corrected if a pilot study is conducted and the lacunae of the survey proforma is corrected . also i think the consensus statement should be different for different regions keeping in mind economic cultural differences
--

REVIEWER	Bilal Alkhaffaf Salford Royal NHS Foundation Trust, Department of Oesophago-Gastric Surgery
REVIEW RETURNED	21-Mar-2022

GENERAL COMMENTS	The authors describe their methodology for developing a COS for primary brain tumours. The protocol describes in detail the different phases of the proposed study which is based on approaches used by previous COS studies and discussed in the COMET handbook. Comments and questions:
---

	1. Scope of COS - the team discuss international applicability, but also confirm that the data used to develop the COS will be primarily UK-based. Have the authors considered international recruitment to the patient interviews, Delphi survey and consensus meeting? 2. Identifying potentially important outcomes. a) Could the authors expand on their justification for using protocols and trial registry entries to identify potentially important outcomes, rather than the widely adopted systematic review of published scientific literature? What would the limitations of this be given that, in my experience from similar studies, trial registries can be notoriously inaccurate and sparse with respect to details on which outcomes are planned to be reported. Whilst there is not a single 'best' way to do this, the team's approach should have its strengths and weaknesses discussed. b) What time period will be covered for the trial registries and qualitative literature review and what is the justification for this? c) The qualitative literature review is an interesting approach to identifying potentially important outcomes. If there is sufficient body of work in this field, what is the need to undertake a further set of interviews which are resource intensive and costly? As things stand, COS are financially costly and take a long time to develop. This approach will be a valuable methodological consideration for future COS developers. d) If interviews are necessary, how many are planned, or predicted to be necessary? e) How will the longlist of outcomes be rationalised into items presented in the Delphi survey? What framework will be used and how many items does the team envisage is an ideal number for participants to prioritise? I look forward to seeing the outcome of this study.
--	--

REVIEWER	Jan Kottner Charité Universitätsmedizin Berlin
REVIEW RETURNED	01-Apr-2022

GENERAL COMMENTS	Thank you very much for the invitation to review this protocol. It follows the COS-STAP statement which can be considered as state-of-the-art. I have the following comments: (1) Introduction, page 3, second para: The sentence “These data are known as...” does not follow from the previous statement. Outcomes are any effects of interventions, not just patient priorities, or clinical benefits. It also includes adverse events and many more. (2) Introduction, page 3, fourth para: Please delete “net benefit”, it is about effects or effectiveness in general. (3) Introduction, page 4, fourth para: What do you mean by “finalise”? Reading the entire protocol, it seems like a new and independent project. If work has already been done, please describe this transparently. (4) Introduction, page 4, fourth para: The aim of COS is to make trial results comparable worldwide. If a COS is going to be developed, it must be done using an international perspective. Focusing on a specific country perspective makes no sense, because we don't want to have country specific COS. The statement in the Dissemination part “... applicability internationally should be
--

	explored...” (page 11) is also very weak. What does “The study team will consider the findings of Existing international initiatives...” mean? If there are international initiatives, they must be involved. (5) Introduction, aim, methods: There is a lot of repetition regarding aims and objectives. The objectives are described in the Background (COS-STAP item 2b) and they don't need to be repeated again and again. The “Objectives” (page 5) belong to the methods and the “Research questions” seems to be little bit out of scope. Especially the last bullet is strange (see above). Later in the methods (page 6) the aims are stated again. List the aims once, the present the methods accordingly. (6) Methods, study design, page 5: Please name COBra first, before using the acronym. (7) Methods, team members, page 5: Please just list who is doing what. Sentences such as “.... underpin the methodological approach...” are strange. Please check whether all abbreviations (MCPCRC etc.) are actually needed later in the text. Please explain what PPI and GRIPP is (page 6). (8) Methods, study summary, page 6: Please delete. (9) Methods, search A, page 7: Please consider to look at published trials too. (10) Methods, stage III, page 9: How exactly will the extracted outcomes classified? My recommendation is to look what was reported and then develop/define the domains inductively and present these outcomes as they are. Classification may be done later. (11) Delphi study: I would recommend to decide later, whether two rounds are sufficient. It really depends on the length of the list and the voting results. If too many outcome are considered critical, then another voting is necessary. (12) Dissemination: Defining COS is good, but please add that outcome measurement instruments need to be developed next. This project will identify the concepts/domains only. (13) Abstract: Please adjust accordingly. Don't say "This paper presents..." Instead describe the objectives.
--	---

VERSION 1 – AUTHOR RESPONSE

Reviewer 1 Glioma outcome trials should have perspective from all the stakeholders . a comprehensive outcome measure would give a better choice of selection and planning for all concerned . this kind of study to measure outcome in a comprehensive way is welcome .	Thank you for your support of this work. Rather than the development or recommendation of an outcome measurement instrument such as a specific questionnaire, this work aims to identify and finalise the core outcomes to be collected and reported in glioma interventional trials, which can be subsequently aligned with measurement tool(kit)s/instruments.
---	---

I suggest the survey questions should be comprehensive and should not be just question number 123 etc but stakeholder should be able to answer q no 8 . there should be provision for missing data which the participant can answer separately .	We recognise that currently used questionnaires / instruments may be burdensome for participants, and mechanisms to address this and minimise missing data are essential. With the development of a core outcome set (COS), the aim is to represent the minimum required outcomes to be collected and reported in glioma studies. This will promote consistency in outcome use rather than volume of data, and may reduce missing data. For the Delphi survey to be conducted in our project, we hope to get input from the participants on all the proposed items, as these are deemed relevant in stages I-III of the project. Of course participants have the option to provide explanations on their choice, including an explanation when they choose not to rate a specific item.
many of the questionnaires fall short on this when the participant does not attempt an answer or the answer is not any of the options . this can be corrected if a pilot study is conducted and the lacunae of the survey proforma is corrected .	Thank you for this comment. In this project we will ensure that the perspective of all stakeholders, including patients, is represented in our work in all stages of the work. Indeed, patients and carers will be included in qualitative interviews, and all stakeholders will take part in the Delphi survey and consensus meeting. In this project, we will provide recommendations on the core outcomes that should be measured, not the specific instruments. The choice for an appropriate instrument should be based on relevance (content validity), but also on other psychometric properties of the instrument, as well as patient burden. This work will be done in the future – this has now been stated in the ethics and dissemination section.
also i think the consensus statment should be different for different regions keeping in mind economic cultural differences	We agree that it is important to consider cross cultural differences. The current COS will be developed from a UK perspective only and aims to include participants from the diverse UK population (i.e. heterogeneous population, reflective of the UK population). For use in other settings or countries, we advise validation exercises to ensure economic and cultural differences are

	integrated. Wording relating to this has been added to the ethics and dissemination section.
Reviewer 2 1. Scope of COS - the team discuss international applicability, but also confirm that the data used to develop the COS will be primarily UK-based. Have the authors considered international recruitment to the patient interviews, Delphi survey and consensus meeting?	Thank you for this comment. Due to the specific experiences and priorities of UK patients within the UK health system, we limited our recruitment to the UK. However, there will be some international participation in the Delphi survey through representation from international regulatory stakeholders. Also, international collaborators are represented on our steering committee and through engagement with colleagues from the Medicines and Healthcare products Regulatory Agency (MHRA), we will be advised on international alignment. Nevertheless, for the COS to be used in other countries / cultures, further validation is necessary. Wording to detail this has been added to the Ethics and Dissemination section.
2. Identifying potentially important outcomes. a) Could the authors expand on their justification for using protocols and trial registry entries to identify potentially important outcomes, rather than the widely adopted systematic review of published scientific literature? What would the limitations of this be given that, in my experience from similar studies, trial registries can be notoriously inaccurate and sparse with respect to details on which outcomes are planned to be reported. Whilst there is not a single 'best' way to do this, the team's approach should have its strengths and weaknesses discussed.	We agree that this is important to discuss and further justification for rationale for using trial registries has therefore been added to the limitations section of the manuscript.
b) What time period will be covered for the trial registries and qualitative literature review and what is the justification for this?	No limitation was placed on the trial registry search and a 15 year limitation was placed on the qualitative literature search. The rationale for this limit was due to the limited data prior to this point and that the literature captured is more reflective of current treatment options and patient perspective. Wording relating to this has been added to the text.
c) The qualitative literature review is an interesting approach to identifying potentially important outcomes. If there is sufficient body of work in this field, what is the need to undertake a further set of interviews which are resource intensive and costly? As things stand, COS are financially costly and take a	Thank you for this insightful observation. This strategy was carefully considered by the steering committee and the study management group. We felt secondary analyses of qualitative literature is limited by the primary interpretation of the data.

long time to develop. This approach will be a valuable methodological consideration for future COS developers.	Including interviews in our project allows in depth exploration of the patient’s perspective and analytical clarity on how this may be reflected in outcomes across the spectrum of glioma. We acknowledge that this is an important question and it is a resource intensive process and not necessarily appropriate for use in all COS. Use of qualitative interviews is encouraged in the COMET (Core Outcome Measures in Effectiveness Trials) handbook and their use was carefully considered in this study. A note relating to this has been added to the strengths and limitations section. We will report on the number of unique outcomes identified from this source to add to evidence base and inform future COS.
d) If interviews are necessary, how many are planned, or predicted to be necessary?	Between 12 and 20 dyads representing the spectrum of malignant disease will be recruited based on previous studies and expected data saturation. Data saturation will be assessed through constant discussion and evaluation of the data by the qualitative researchers conducting the data collection and analysis, together with members of the wider study team. Recruitment will end when data saturation is reached. This is currently reported in the methods section.
e) How will the longlist of outcomes be rationalised into items presented in the Delphi survey? What framework will be used and how many items does the team envisage is an ideal number for participants to prioritise?	Details relating to the development of the outcomes lists are now more extensively reported in stage III of the methods, and is in accordance with the approach outlined in the COMET handbook.
Reviewer 3 (1) Introduction, page 3, second para: The sentence “These data are known as...” does not follow from the previous statement. Outcomes are any effects of interventions, not just patient priorities, or clinical benefits. It also includes adverse events and many more .	Following the reviewer’s suggestion, this sentence has been re-phrased and linked with the next paragraph.
(2) Introduction, page 3, fourth para: Please delete “net benefit”, it is about effects or effectiveness in general.	The word “net” has been removed.
(3) Introduction, page 4, fourth para: What do you mean by “finalise”? Reading the entire protocol, it seems like a new and independent project. If work	This was indeed not phrased clearly. We have now clarified that this is a new independent project where a COS will be

has already been done, please describe this transparently.	developed and finalised in a consensus meeting.
(4) Introduction, page 4, fourth para: The aim of COS is to make trial results comparable worldwide. If a COS is going to be developed, it must be done using an international perspective. Focusing on a specific country perspective makes no sense, because we don't want to have country specific COS. The statement in the Dissemination part "... applicability internationally should be explored..." (page 11) is also very weak. What does "The study team will consider the findings of Existing international initiatives..." mean? If there are international initiatives, they must be involved.	Thank you for this feedback. There are three sources of data used in the identification of outcomes – the trial registries, qualitative literature, and qualitative interviews. The trial registry searches were without restriction based on country and the qualitative literature was limited to those in English language only. Both of these can be considered as international sources. However, the qualitative interviews were only undertaken in the UK so their experiences and priorities may be specific to and shaped by the UK health system. The study steering committee has membership stakeholders leading international initiatives and the Delphi survey and consensus meeting will involve participants from international regulatory bodies. The statement in the introduction describing the UK perspective has been removed and the statement in the dissemination section has been re-phrased. This has also been re-worded in the "Focus of COS" section. Nevertheless, for the COS to be used in other countries / cultures, further validation is necessary. Wording relating to this has now been added to the ethics and dissemination section.
(5) Introduction, aim, methods: There is a lot of repetition regarding aims and objectives. The objectives are described in the Background (COS-STAP item 2b) and they don't need to be repeated again and again. The "Objectives" (page 5) belong to the methods and the "Research questions" seems to be little bit out of scope. Especially the last bullet is strange (see above). Later in the methods (page 6) the aims are stated again. List the aims once, the present the methods accordingly.	The aims and objectives section has been split out so the aims are included in the background section and the repetitive parts are removed. The objectives have been moved to the methods section. The research questions have been removed. We hope that the adjusted manuscript reads better.
(6) Methods, study design, page 5: Please name COBra first, before using the acronym.	Thank you for noticing – this has now been corrected.
(7) Methods, team members, page 5: Please just list who is doing what. Sentences such as ".... underpin the methodological approach..." are strange. Please check whether all abbreviations (MCPCRC etc.) are actually needed later in the text. Please explain what PPI and GRIPP is (page 6).	Thank you – this has been re-worded to clarify the role. The abbreviations that are not used again in the manuscript have been removed and GRIPP2 has been provided in full. PPI is given in full in the first section of the study team section.

(8) Methods, study summary, page 6: Please delete.	Following the reviewer's suggestion, this has been deleted.
(9) Methods, search A, page 7: Please consider to look at published trials too.	Given the under-reporting of outcomes in trial publications, we decided to use trial registries instead. This issue was also raised by reviewer 1 and further information for the rationale and exploration of the limitations of this approach have been added to the manuscript.
(10) Methods, stage III, page 9: How exactly will the extracted outcomes classified? My recommendation is to look what was reported and then develop/define the domains inductively and present these outcomes as they are. Classification may be done later.	The approach for this has been provided in more detail, outlining the development of an ontology and its use as a categorical tool.
(11) Delphi study: I would recommend to decide later, whether two rounds are sufficient. It really depends on the length of the list and the voting results. If too many outcome are considered critical, then another voting is necessary.	Thank you for raising this issue, which we have carefully discussed when setting up the study. The decision to proceed with two rounds was determined in advance as the details of participation are required for informed consent of participants. Two rounds were decided to mitigate attrition between rounds and minimise missing data, reduce time required of participants and promote data completeness. Further to this, the study team was conscious of participant burden in this particular population. The decision reflects the view of the steering committee. The consensus meeting allows for a further opportunity to discuss, validate and confirm the final. Wording to this effect has been added to the consensus meeting section of the methods.
(12) Dissemination: Defining COS is good, but please add that outcome measurement instruments need to be developed next. This project will identify the concepts/domains only.	We agree that it is important to emphasize that only the outcomes are identified, and that instruments to assess these outcomes (appropriately) should be identified in later stages. Wording to this effect has now been added to the ethics and dissemination section.
(13) Abstract: Please adjust accordingly. Don't say "This paper presents..." Instead describe the objectives.	Thank you, this has now been re-worded.

VERSION 2 – REVIEW

REVIEWER	Bilal Alkhaffaf Salford Royal NHS Foundation Trust, Department of Oesophago-Gastric Surgery
REVIEW RETURNED	19-May-2022

GENERAL COMMENTS	I am satisfied with the responses to my comments. I have no further questions for the authors and wish them the best of luck in their project.
--

REVIEWER	Jan Kottner Charité Universitätsmedizin Berlin
REVIEW RETURNED	14-May-2022

GENERAL COMMENTS	Thank you very much for the invitation to review this COS development protocol again. It has been improved. I have the following final comments: (1) Still I'm not convinced about the justification about the UK perspective. The idea of COS is to standardize trial outcomes globally. Country-specific COS are not helpful. Stage I, search strategy (page 7): It says that the trial registries will be searched 'based in the US and UK'. Does this mean that trial conducted in other countries will be ignored? (2) Objectives (page 6): The numbers start with 5.
---

VERSION 2 – AUTHOR RESPONSE

Reviewer comment	Author response
Reviewer 3 Thank you very much for the invitation to review this COS development protocol again. It has been improved.	Thank you for this positive feedback.
(1) Still I'm not convinced about the justification about the UK perspective. The idea of COS is to standardize trial outcomes globally. Country-specific COS are not helpful. Stage I, search strategy (page 7): It says that the trial registries will be searched 'based in the US and UK'. Does this mean that trial conducted in other countries will be ignored?	The wording around this has been adjusted further in the discussion and references to the UK perspective has now been removed. Our intention was to be forthright about the possible limitations related to drawing on a UK sample for the original qualitative component and a largely UK population for the Delphi survey. However, the other data sources are without restriction based on country (though the registries are based in the UK and US), and the qualitative systematic review is only limited by English language. The development of the COS will be inclusive of all glioma trials meeting our criteria and not just UK trials. We do state though that there may be some cultural/economic validation required for use with other populations, as is the

	recommendation for most COS.
(2) Objectives (page 6): The numbers start with 5.	Thank you, this is now correct.
Reviewer 2 I am satisfied with the responses to my comments. I have no further questions for the authors and wish them the best of luck in their project.	Thank you for this kind feedback.